# SpaceByte: Towards Deleting Tokenization from Large Language Modeling

**Kevin Slagle**
Rice University
`kevin.slagle@rice.edu`

## Abstract

Tokenization is widely used in large language models because it significantly improves performance. However, tokenization imposes several disadvantages, such as performance biases, increased adversarial vulnerability, decreased character-level modeling performance, and increased modeling complexity. To address these disadvantages without sacrificing performance, we propose SpaceByte, a novel byte-level decoder architecture that closes the performance gap between byte-level and subword autoregressive language modeling. SpaceByte consists of a byte-level Transformer model, but with extra larger transformer blocks inserted in the middle of the layers. We find that performance is significantly improved by applying these larger blocks only after certain bytes, such as space characters, which typically denote word boundaries. Our experiments show that for a fixed training and inference compute budget, SpaceByte outperforms other byte-level architectures and roughly matches the performance of tokenized Transformer architectures.

## 1 Introduction

Most language models are trained using tokenization, which partitions text into tokens that typically consist of words or subwords. Tokenization is useful because it significantly decreases the inference and training computational costs of large language models. However, tokenization also imposes several disadvantages, including performance penalties for languages that are priortizes less by the tokenizer [1–3]; increased vulnerability to adversarial attacks [4]; and worse character-level modeling performance [5, 6], and additional model complexity.[1]

Recently, MegaByte [7], MambaByte [6], and more [8–12] have been proposed as new byte-level autoregressive language models that model bytes instead of tokens. (See [13–21] for encoder and encoder-decoder byte-level modeling.) To address the longer context size resulting from modeling bytes instead of tokens, MegaByte uses multiscale modeling [22–24], while MambaByte uses Mamba blocks [25] instead of Transformer blocks. But although MegaByte and MambaByte have been shown to perform better than a standard byte-level Transformer, to our knowledge, no byte-level autoregressive large language model architecture has been shown to match the performance of tokenized models when controlling for compute costs.

In this work, we study the performance of byte-level and subword-level autoregressive models when trained using a fixed compute budget. We measure the performance in terms of the cross entropy (measured in bits-per-byte), which has been shown to be a strong predictor of down-stream performance [26]. In addition to controlling for training compute, we also control for inference compute costs (measured in FLOPs). We find that byte-level Transformer and MegaByte models can require roughly 10 times more training FLOPs to achieve the same performance as a subword-

---

[1]See also Andrej Karpathy's tweet twitter.com/karpathy/status/1657949234535211009 and video youtube.com/watch?v=zduSFxRajkE&t=6725s on the disadvantages of tokenization.

38th Conference on Neural Information Processing Systems (NeurIPS 2024).

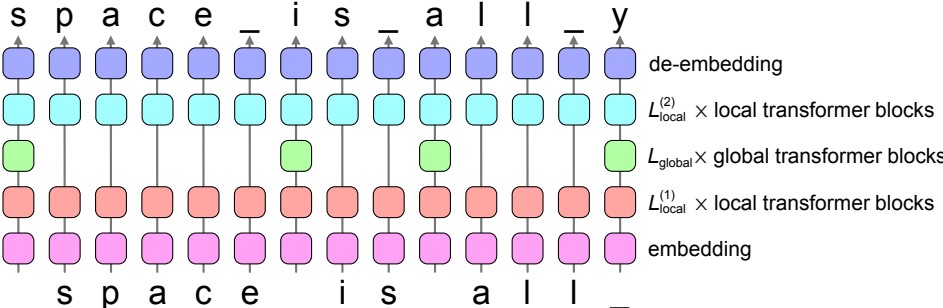

Figure 1: An overview of the SpaceByte architecture. The embedding, local transformer blocks, and de-embedding (i.e. a layer norm and linear) are the standard Transformer decoder layers. SpaceByte modifies the standard transformer by applying "global" transformer blocks only after certain bytes, such as space characters. The intuition is that the first character of a word is typically the hardest to predict; thus this positioning of the global blocks should make the best use of the global blocks (which use a larger model dimension).

level Transformer. To close this substantial performance gap, we propose a new byte-level decoder architecture: SpaceByte.

SpaceByte also utilizes multiscale modeling to improve efficiency by grouping bytes into patches. But unlike MegaByte, which uses a fixed patch size, SpaceByte uses a simple rule to dynamically partition the bytes into patches that are aligned with word and other language boundaries. Our compute-controlled experiments show that this simple modification is crucial for performance, allowing SpaceByte to outperform other byte-level architectures and roughly match the performance of subword Transformers across a variety of text modalities.

Our experiments are performed on datasets consisting of English books, LaTeX formatted arXiv papers, and open-source code. For other data modalities, SpaceByte with our simple patching rule might not be as effective.

## 2   SpaceByte

The SpaceByte architecture is summarized in Figure 1. In a nutshell, SpaceByte can be thought of as a byte-level Transformer model, but with extra "global" transformer blocks (with a larger model dimension) inserted in the middle, which are only applied a fraction of the time. While the MegaByte architecture applies the global transformer blocks every $P \sim 8$ bytes, we hypothesize that this fixed spacing hinders performance. Our intuition is that the first character of a word is typically significantly harder to predict than the following characters. We therefore expect that performance can be improved by applying the global blocks primarily at word boundaries.

**Global Block Insertion Rule** In this work, we consider a very simple rule to dynamically decide when to apply the global blocks. We assume that the text bytes are encoded using the UTF-8 encoding. We define a byte to be *spacelike* if the byte does not encode a letter, number, or UTF-8 continuation byte[2]. We apply the global blocks after any spacelike byte that is not preceded by another spacelike byte (and after any BOS token). See Figure 2 for examples.

The most common spacelike byte is the space character. Thus, the global blocks are applied most frequently to predict the first character of a word, which we expect is the hardest character to predict [27] in a given word. With fixed patch size (e.g. as in MegaByte), the global blocks are typically inserted in the middle a word, which we expect is inefficient because predicting the rest of the word could likely be more efficiently accomplished using the local blocks. We define continuation bytes to be spacelike so that languages that do not use spaces between words can still benefit from the global blocks between multi-byte characters (e.g. Chinese characters consists of three bytes in UTF-8).

---

[2]UTF-8 uses a variable number of bytes to encode a character. English letters or numbers consist of a single byte. Characters that are encoded using multiple bytes are encoded using a leading byte (which is spacelike by our definition) followed by one or more continuation bytes (which are not spacelike).

PG-19:

```
the↓enemy!''●●_he_exclaimed._``●●Their_capture_must_be_prevented._Come_with_
```

arXiv:

```
where_$q_1=q_2=\dots=q_\kappa$_and_$V_1=V_2=\dots_V_\kappa$._In_this_way,
```

Github:

```
____exp_+=_2;↓↓____mbf[3]_=_exp;↓____mbf[2]_=_sign_|_(ieee[2]_&_0x7f);↓
```

Figure 2: Examples of patch boundaries from datasets that we study. Spacelike bytes are underlined and colored blue. Patches boundaries are drawn above the text. Each patch ends after a spacelike byte that is not preceded by another spacelike byte. Consequently, each patch begins with zero or more spacelike bytes, followed by one or more non-spacelike bytes, and ends with a single spacelike byte. The global blocks predict the first character of each patch. The downward arrow ($\downarrow$) denotes a newline byte. The left and right quotation characters, (") and (") in the PG-19 example, are encoded using three bytes in UTF-8. The first of the three bytes is spacelike, while the later two bytes are UTF-8 continuation bytes, which are not spacelike and are each denoted using a bullet point (•) above.

Although this very simple "spacelike" rule is likely not the optimal rule, we find that it works surprisingly well in practice for English text, LaTeX formatted papers, and code. Nevertheless, a critical future direction is to optimize [28, 14, 9] better rules using data rather than our simple heuristic.

**Important Details** Since the global blocks are not applied as often as the local transformer blocks, it is advantageous to use a larger model dimension for the global transformer blocks. To increase the dimensions of an activation vector before the global blocks, we simply pad the activation vector with zeros. To decrease the dimension, we truncate the activation vector.

In our experiments, we use a significantly larger context size than the model dimension $D_{\text{local}}$ of the local transformer blocks. To prevent the attention mechanism from dominating the compute costs for the local model, we use an attention window [29–31] of length $D_{\text{local}}$ for the local transformer blocks. The global blocks use a global attention that attends to all other global blocks.

See Appendix C for pseudocode. Additional details specific to our experiments are provided in Sections 4.1 and 4.2 and Appendix A.

## 3 Related Work

The most straight-forward consequence of modeling bytes instead of subword tokens is that the length of a sequence typically increases by about a factor of four. This increased sequence length increases the training and inference compute cost for modeling a given long sequence of text for a Transformer due to the quadratic scaling of attention.

**MegaByte** The MegaByte architecture strives to use multiscale Transformer modeling to lessen these performance issues. In particular, MegaByte groups bytes into patches of a fixed patch size $P$. Each patch of bytes is vectorized and then fed into a "global" Transformer model. The output of the global model is then fed into a "local" Transformer model that autoregressively outputs byte-level logits. [7]

For a context size of $T$ bytes, MegaByte's global Transformer model compresses the context into only $T/P$ patches, which can significantly decrease the compute cost for modeling long sequences. Similar to Yu et al. [7], we also find that MegaByte outperforms a standard byte-level Transformer. However, we find that MegaByte's performance is remarkably close to a stronger byte-level Transformer baseline that simply uses a sliding window attention mechanism [29–31] to increase the context size without increasing the compute costs.

Yu et al. [7] do not compare MegaByte to subword-level Transformer in compute controlled experiments. In our compute controlled experiments, we find that MegaByte's performance significantly lags behind a subword-level Transformer.

Compared to MegaByte, SpaceByte makes the crucial change that patches are dynamically sized to be commensurate with the text, e.g. with word boundaries. We also add an additional local model before the global model (while MegaByte only utilizes a single local model after the global model) to help the model deal with the dynamical patch sizes. We also use significantly longer attention windows for our local models. We find that these changes allow SpaceByte to significantly improve upon the performance of MegaByte and roughly match the performance of subword-level Transformers.

**MambaByte** The MambaByte architecture [6] takes an alternative approach to avoiding the quadratic compute scaling of the attention mechanism by replacing the Transformer block with a Mamba block [25]. Wang et al. [6] find that their byte-level MambaByte models outperform byte-level Transformer and byte-level MegaByte models. They perform one controlled experiment with a subword model, where they find that MambaByte slightly outperforms Mamba (using tokens) when controlling for the amount of model parameters and training data (which was 14 epochs of the PG-19 dataset). But this experiment was not controlled for compute as MambaByte was trained using roughly four times as much compute than Mamba. We view the Mamba and MambaByte architectures as complementary to our work, as the Mamba block could be integrated into SpaceByte (or MegaByte) in place of Transformer blocks.

**Layer Skipping** SpaceByte could be though of as a Transformer that employs a novel kind of text-dependent layer skipping [32–38] on the middle layers.

**Word Boundary** Prior works have shown utility in using word boundaries to partition patches for autoregressive multi-scale byte-level modeling [9, 8, 11] (and also [18] for encoder-decoder modeling). However, these works did not compare autoregressive byte-level models to subword-level models, nor did they identity a patch partitioning rule that could generically be applied to UTF-8 encoded text. Our primary contributions beyond these prior works is to show how to scale word-boundary byte-level modeling to more diverse text modalities while roughly matching the performance of subword-level models in compute-controlled experiments.

Nawrot et al. [9] and Fleshman and Van Durme [11] make use of the Hourglass Transformer architecture [23]. The SpaceByte architecture is similar to the Hourglass Transformer, except SpaceByte uses a simpler technique for shortening and upscaling the activations before and after the global blocks, and SpaceByte uses a sliding window attention [29–31] in the local blocks to improve performance for long context sizes.

## 4    Experiment Setup

Our experiments compare the performance of our byte-level SpaceByte architecture to subword-level Transformer and byte-level Transformer and MegaByte architectures. To fairly compare the performance between the byte and subword level models, we measure the cross-entropy of the test dataset in terms of bits-per-byte.[3] Given the substantial variation in inference compute costs across the models we study, we also compare their inference compute costs to provide a more comprehensive evaluation. We use FLOPs-per-byte as a simple software and hardware–independent proxy for inference compute costs, which is the number of FLOPs (see Appendix A.1) required to model a byte of text.[4]

Note that by controlling for both total training compute and FLOPs-per-byte, we are also controlling for the amount of training data since (bytes trained) = (training FLOPs)/(training FLOPs-per-byte). The FLOPs-per-byte during training is equal to three times the FLOPs-per-byte during inference (due to the backwards pass during training).

---

[3]The bits-per-byte (BPB) is equal to $\text{BPB} = \text{XE} \, N_{\text{tokens}}/(N_{\text{bytes}} \ln 2)$, i.e. the cross entropy per token (XE) times the number of tokens per byte ($N_{\text{tokens}}/N_{\text{bytes}}$) divided by $\ln 2$ (to convert nats to bits). $N_{\text{tokens}}$ and $N_{\text{bytes}}$ are the number of tokens and bytes in the dataset, respectively. For byte-level models, $N_{\text{tokens}} = N_{\text{bytes}}$ since bytes are used in place of tokens.

[4]We note that in order for memory bandwidth to not be a bottleneck during inference, the batch size must be sufficiently large and e.g. grouped-query attention [39, 40] must be used.

We therefore study the Pareto frontier of lowest bits-per-byte and lowest FLOPs-per-byte. We train all models using a compute-controlled setup, using either $10^{18}$ or $10^{19}$ FLOPs. In order to effectively explore this Pareto frontier, we train models using a grid of different model dimensions and numbers of layers, as specified in Appendix B.3.

**Datasets** Following the MegaByte [7] and MambaByte [6] experiments, we benchmark our models on a diverse range of long-form datasets: PG-19 (English-language books written before 1919) [41]; arXiv (papers from ArXiv written in LaTeX, extracted from the arXiv component of The Pile [42]); and Github (open-source code repositories, extracted from the Github component of The Pile [42]).

## 4.1 Models

The models we study tend to perform best when the compute cost is roughly evenly split between the attention and feedforward layers. To ensure this, we fix the context size (or attention window) to be equal to the model dimension for every layer. We detail our model setup below.

**Notation** For all models, we use $T$ to denote the context length, and $D$ to be the model dimension (of the global model for SpaceByte and MegaByte).

For SpaceByte and MegaByte, $D_{\text{local}}$ is the dimension of the local model, and $T_{\text{global}}$ is the maximum context size for the global model. The patch size $P$ is the number of bytes between global blocks. If the patch size is fixed (which is always the case for MegaByte), we naturally set the context size to be $T = PT_{\text{global}}$.

Below, we describe each of the model architectures that we compare in our experiments.

**SpaceByte** We fix the global context size and global model dimension to be equal, $T_{\text{global}} = D$, and we set the local attention window $W_{\text{local}}$ equal to the local model dimension, $W_{\text{local}} = D_{\text{local}}$. For the PG-19 and arXiv datasets, the average patch size is roughly 6, so we take $T = 6T_{\text{global}}$ for these datasets; for the Github dataset, the average patch size is roughly 8, so we instead take $T = 8T_{\text{global}}$ for the Github dataset.

For simplicity, we fix the number of global transformer blocks $L_{\text{global}}$ to be equal to the total number of local blocks, $L_{\text{local}}^{(1)} + L_{\text{local}}^{(2)}$, and we evenly split the number of local blocks before ($L_{\text{local}}^{(1)}$) and after ($L_{\text{local}}^{(2)}$) the global blocks, i.e. we fix $L_{\text{local}}^{(1)} = L_{\text{local}}^{(2)} = \frac{1}{2}L_{\text{global}}$.

**SpaceByte (fixed patches)** To clearly demonstrate the utility of dynamically aligning patch boundaries in SpaceByte, we also train a simplified version SpaceByte where the patches all have a fixed size. In order to roughly match SpaceByte's average patch size, we take the fixed patch size to be $P = 6$ for all datasets except for the Github dataset, for which we use $P = 8$. We again use $T_{\text{global}} = D$ and $T = PT_{\text{global}}$.

**Byte-level Transformer** For a simple baseline comparison (following Yu et al. [7]), we train byte-level Transformer models. We take the context size to be equal to the model dimension, $T = D$.

Note that in our setup, a Transformer with model dimension $D$ only sees a context size of $D$, which is significantly smaller than the context size of $PD$ for SpaceByte (and MegaByte) with patch size $P$.

**Byte-level Transformer (Window Attention)** Since a shorter context is a significant disadvantage for long-form datasets, we also compare against a stronger Transformer baseline that uses a sliding window attention [29–31] to efficiently increase the context size without increasing compute costs. We train each window attention enhanced Transformer using a context size $T = PD$ and a sliding attention window size equal to $D$, with $P = 6$ for all datasets except for the Github dataset for which $P = 8$.[5]

**MegaByte** We also compare to MegaByte [7]. Although MegaByte was originally trained using a patch size of $P = 8$, we found that a patch size of $P = 4$ was often better in our setup. We thus include both of these patch sizes (4 and 8) in our hyperparameter grid for MegaByte. For simplicity, we fix the number of layers in the global and local blocks to be equal, $L_{\text{global}} = L_{\text{local}}$, which is close

---

[5]We also tried a simplified Sparse Transformer [30] where the first attention layer uses a sliding attention window of size $D$; the second attention layer uses a strided attention with stride $P$; and the remaining layers continue to alternate between sliding and strided attention. However in our setup, we found this to perform worse than just using a sliding window attention.

Table 1: **Best bits-per-byte.** Lowest bits-per-byte[3] for each model architecture when trained using $10^{19}$ FLOPs on different text modalities. The lowest bits-per-byte for each dataset are underlined; and the lowest within 2.5% are bolded. The largest statistical error (due to a finite number of evaluation samples) is 0.4%. SpaceByte significantly outperforms other byte-level architectures and performs on par with the SentencePiece subword Transformer.

| | Model | PG-19 | arXiv | Github |
|---|---|---|---|---|
| subword | Transformer (GPT2 tokenizer) | 1.013 | 0.796 | 0.554 |
| | Transformer (SentencePiece) | **0.989** | 0.768 | **0.508** |
| byte-level | Transformer | 1.138 | 0.909 | 0.655 |
| | Transformer (Window Attention) | 1.089 | 0.818 | 0.560 |
| | MegaByte | 1.083 | 0.822 | 0.570 |
| | SpaceByte (fixed $P$) | 1.112 | 0.804 | 0.552 |
| | SpaceByte | **1.009** | **0.748** | **0.500** |

to what was used by Yu et al. [7]. Similar to SpaceByte, we set the context size to $T = PD$, where $D$ is the global model dimension.

**Subword Transformers** Our most important baseline is the standard subword Transformer. We train subword Transformers using two different tokenizers (both with a vocabulary size of 50,257): (1) the GPT2 tokenizer [43], and (2) a SentencePiece [44] tokenizer using a byte-pair-encoding model [45] that was separately trained for each dataset. As usual, we set the context size to be equal to the model dimension, $T = D$.

### 4.2 More Details

We use fairly standard Pre-LN [46, 30, 47] Transformer [48] blocks with no bias terms. Since MegaByte uses Rotary Position Embedding (RoPE) [49], we also use RoPE for all models (which slightly improves the loss). To prevent loss divergences during training, we use qk-layernorm [50–52] (which we strongly recommend) for all models; i.e. we add an extra layer-normalization to the query and key vectors in the self-attention layers.

All hyperparameters have been carefully tuned using grid and random searches. See Appendices A and B for more details.[6]

## 5 Results

We now present our experimental data comparing the different model architectures in compute-controlled settings. Figure 3 plots the Pareto frontier of lowest cross-entropy bits-per-byte and lowest FLOPs-per-byte (i.e. inference compute cost) for each architecture and training compute budget. We assume that the Pareto frontier is convex. Thus, for each architecture and compute budget, we perform a grid search over model dimension and number of layers; we then draw a piecewise-linear line connecting the best (i.e. minimal subset of) models such that all other models (not shown in figure) lie above and to the right of the line. Table 1 summarizes the results for the lowest overall bits-per-byte for each architecture.

Across all datasets, training compute budgets, and inference compute budgets (i.e. FLOPs-per-byte), SpaceByte significantly outperforms all other byte-level architectures. SpaceByte also consistently outperforms the subword Transformer when using GPT2 tokens, and by a wide margin on the arXiv and Github datasets. SpaceByte roughly matches the performance of the most competitive baseline, the subword Transformer using the SentencePiece tokenizer, with SpaceByte performing slightly better on the arXiv and Github datasets. Figure 3 also suggests that SpaceByte's performance improves faster than the subword Transformer as the training compute budget increases.

Byte-level architectures other than SpaceByte perform significantly worse than SpaceByte or the SentencePiece Transformer. For example, for PG-19, the next best byte-level architecture is

---

[6]Our training code and data reproduction steps can be found at github.com/kjslag/spacebyte.

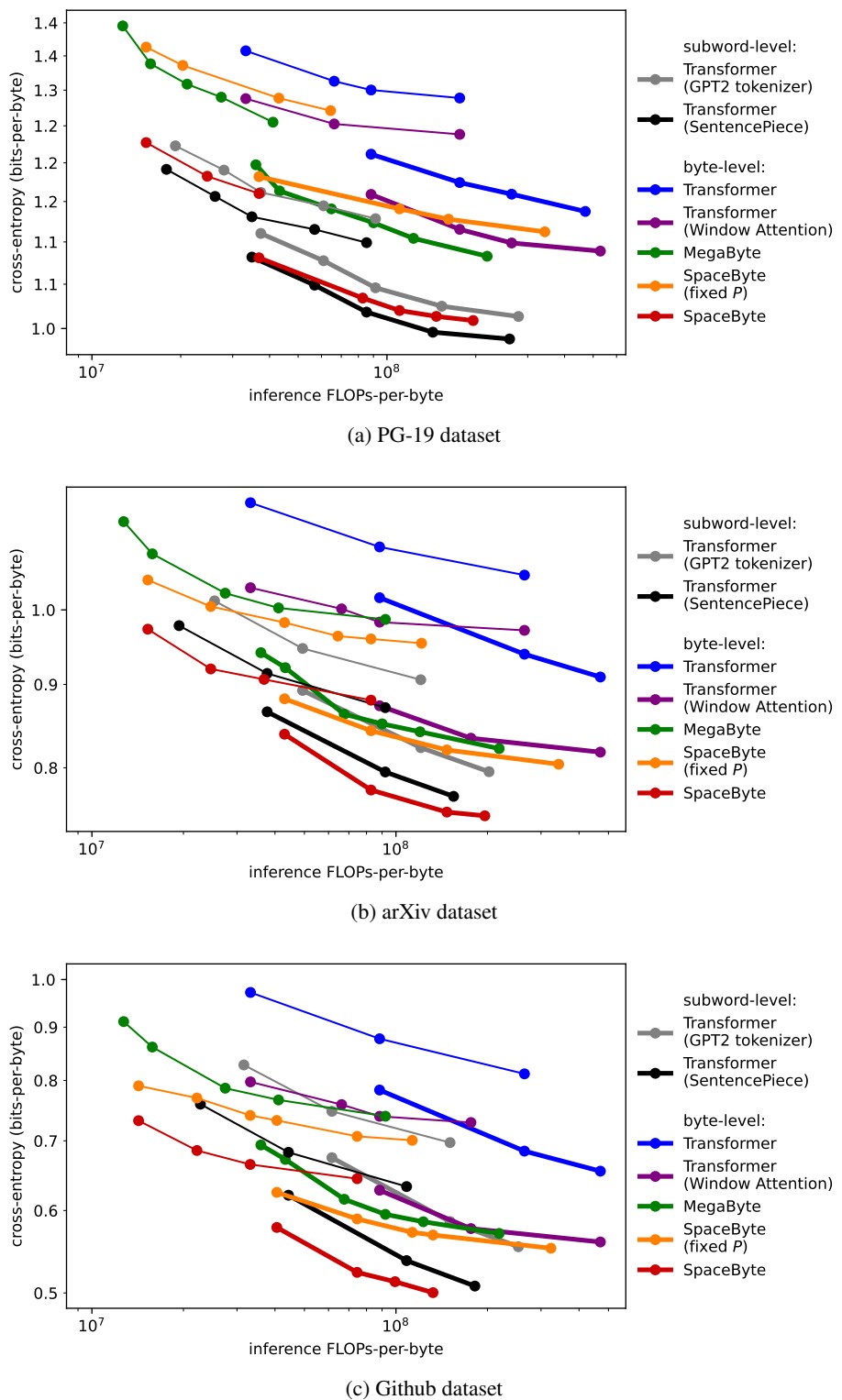

(a) PG-19 dataset

(b) arXiv dataset

(c) Github dataset

Figure 3: Pareto frontier of the cross-entropy bits-per-byte[3] vs FLOPs-per-byte during inference (details in Appendix A.1) for each model architecture trained using $10^{18}$ (connected by thin lines) or $10^{19}$ (thick lines) FLOPs on different datasets (on a log-log scale). Each dot describes a model with a different number of layers and/or model dimension. Lower and to the left is better. SpaceByte (red) outperforms all other byte-level architectures across the entire Pareto frontier for all datasets. SpaceByte roughly matches the performance of the subword Transformer using SentencePiece tokens, and outperforms the subword Transformer using GPT2 tokens.

Table 2: **Comparison with other works.** We compare SpaceByte to byte-level models trained in other works, along with a subword transformer that we train. All models are trained using roughly the same inference FLOPs-per-byte ($\approx 728$M). The bits-per-byte for the Transformer, PerceiverAR, and MegaByte models are taken from Yu et al. [7], while MambaByte results are taken from Wang et al. [6]. The best bits-per-byte for each dataset are underlined; and the lowest within 3% are bolded. The largest 1-sigma statistical error (due to a finite number of evaluation samples) for the models we train is less than 0.001. SpaceByte is the overall best performing byte-level model and consistently performs within a few percent of the subword Transformer.
† These models used slightly different datasets for training and/or testing. For MambaByte-353M, we estimate that this difference very roughly amounts to an extra 3% statistical error.

| | Model | Context size | Data trained | Test bits-per-byte ↓ | | | |
| --- | --- | --- | --- | --- | --- | --- | --- |
| | | | | PG-19 | Stories | arXiv | Github |
| subword | Transformer-1B | 2048 tokens $\sim 8192$ bytes | $\approx 30$B* bytes | **0.908** | **0.809** | 0.666 | **0.400** |
| byte-level | Transformer-320M [7] | 1024 | 80B | 1.057 | 1.064 | 0.816† | 0.575† |
| | PerceiverAR-248M [7] | 8192 | 80B | 1.104 | 1.070 | 0.791† | 0.546† |
| | MegaByte-758M+262M [7] | 8192 | 80B | 1.000 | 0.978 | **0.678**† | **0.411**† |
| | MambaByte-353M [6] | 8192 | 30B* | **0.930** | **0.908**† | **0.663**† | **0.396**† |
| | SpaceByte-793M+184M | 8192 (bytes) | 30B* (bytes) | **0.918** | **0.833** | **0.663** | **0.411** |

MegaByte; however, MegaByte trained using $10^{19}$ FLOPs (thick green line in Figure 3a) performs worse across nearly the entire Pareto frontier than the SentencePiece Transformer trained using only 10% as many training FLOPs (thin black line). Although the standard byte-level transformer (which is the primary baseline used by Yu et al. [7], blue in Figure 3) performs significantly worse than the other byte-level models, we note that by simply using a sliding window attention mechanism to increase the context size to more closely match that of the other byte-level models, this stronger baseline (purple) performs almost as well as MegaByte. Nevertheless, SpaceByte still significantly outperforms this stronger baseline.

To verify the importance of dynamic patch sizes for SpaceByte's performance, we compare SpaceByte to a variant of SpaceByte with fixed patch sizes (orange in Figure 3). We observe that fixing the patch size significantly degrades the performance of SpaceByte.

Note that on the arXiv and Github datasets, the subword Transformer performs significantly worse when using GPT2 tokens (which were trained on WebText [43]) than SentencePiece tokens (which were trained using the specific dataset). This exemplifies the bias that tokenization can introduce on data distributions different from what the tokenizer was trained on.

## 6 Comparison with Other Works

We also compare SpaceByte performance to byte-level models trained in other works. Yu et al. [7] trained Transformer, PerceiverAR, and MegaByte models, each using the same amount of compute, FLOPs-per-byte, and data (80B bytes). Wang et al. [6] additionally trained a MambaByte model using the same FLOPs-per-byte but only 30B bytes of data. We train SpaceByte-793M+184M ($D = 1536$, $D_{\text{local}} = 768$, $L_{\text{local}} = 26$, $L_{\text{global}} = 28$) using roughly the same inference FLOPs-per-byte (728M) but also only 30B bytes of data (following Wang et al. [6]). Training these models thus requires roughly $3 \times 728$M FLOPs-per-byte $\times$ 30B bytes $\approx 6.5 \times 10^{19}$ FLOPS, where the factor of three comes from converting inference FLOPs-per-byte to training FLOPs-per-byte (which additionally requires a backwards pass). For this experiment, we set the context size of SpaceByte to 8192 bytes to follow the prior works. See Appendix A for more details.

We also train subword Transformer-1B ($D = 1536$) models using the SentencePiece tokenizer (except for the Stories dataset, for which we use the GPT2 tokenizer). The average number of bytes per token

for the PG-19, Stories, arXiv, and Github datasets are 4.05, 4.39, 3.73, and 3.31, respectively. To match the FLOPs-per-byte of the subword Transformer-1B models to the byte-level models, we set the number of layers to 40, 44, 37, or 31, for Transformer-1B on these four respective datasets.

Results are shown in Table 2. We show experiments for the PG-19 [41], Stories [53], arXiv (extracted from The Pile [42]), and Github (extracted from The Pile [42]) datasets.[7] Yu et al. [7] used proprietary "arXiv" and "Code" datasets, which we do not have access to. Following Wang et al. [6], we compare Yu et al. [7]'s results to the similar (but likely slightly different) arXiv and Github components of The Pile [42]. However, Wang et al. [6] use their own test splits to evaluate MambaByte-353M on Stories, arXiv, and Github. Due to the rather small test splits ($\sim 100$MB for the arXiv and Github datasets), this difference can be significant. For example, the validation (and test) bits-per-byte for SpaceByte-793M+184M on the Stories, arXiv, and Github datasets are 0.877 (0.833), 0.658 (0.663) and 0.397 (0.411), which differ by $+5\%$, $-1\%$, and $-3\%$, respectively. Given this variation, the bits-per-byte of MambaByte-353M and SpaceByte-793M+184M are not statistically different on the arXiv or Github datasets.

Overall, we find that SpaceByte outperforms the byte-level models trained in other works. SpaceByte outperforms MegaByte, even though MegaByte was trained using 2.7 times as much compute and data. Moreover, SpaceByte's performance is competitive with the subword Transformer-1B.

## 7    Conclusion

We have proposed a new byte-level Transformer decoder architecture, SpaceByte. Our compute-controlled experiments show that SpaceByte outperforms all other byte-level architectures and roughly matches the performance of sub-word level Transformers.

**Limitations** SpaceByte uses a simple byte partitioning rule that relies on "spacelike" bytes, such as spaces which typically denote word boundaries. As such, SpaceByte should not be expected to perform well on arbitrary sequences of bytes, such as images or audio. Some languages, such as Chinese, do not use spaces between words. SpaceByte is somewhat robust to these languages, since e.g. Chinese characters are encoded using three bytes in UTF-8, which SpaceByte will group together. However, our preliminary experiments suggest that SpaceByte performs worse than subword transformers on Chinese text. It would therefore be desirable to improve upon and generalize SpaceByte's global block insertion rule.

The variable spacing between global blocks makes it more challenging to design and implement an efficient batched inference sampling algorithm for SpaceByte.

**Future Work** SpaceByte uses multiscale modeling where the local model operates on bytes while the global model typically operates on words. Another natural extension of our work is to try recursively applying multiscale modeling at even longer scales, such as the sentence or paragraph level. It would also be fruitful to investigate if Mamba blocks [25] could further improve SpaceByte's performance.

## Acknowledgments and Disclosure of Funding

We thank Tushaar Gangavarapu, Junxiong Wang, and Lili Yu for helpful conversations. This work was supported in part by the NSF Campus Cyberinfrastructure grant CC* Compute: Interactive Data Analysis Platform OAC-2019007 and by Rice University's Center for Research Computing (CRC).

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

Table 3: **Model hyperparameters.** Hyperparameters for models shown in Table 1.

| Model | Dataset | Parameters | FLOPs-per-byte | $L$ $(L_\text{global}/L_\text{local})$ | $D$ $(D/D_\text{local})$ | $T$ |
|---|---|---|---|---|---|---|
| Transformer (GPT2 tokenizer) | PG-19 | 454M | 279M | 32 | 1024 | 1024 |
| | arXiv | 253M | 202M | 16 | 1024 | 1024 |
| | Github | 253M | 253M | 16 | 1024 | 1024 |
| Transformer (SentencePiece) | PG-19 | 454M | 260M | 32 | 1024 | 1024 |
| | arXiv | 253M | 155M | 16 | 1024 | 1024 |
| | Github | 253M | 182M | 16 | 1024 | 1024 |
| Transformer | PG-19 | 202M | 470M | 16 | 1024 | 1024 |
| | arXiv | 202M | 470M | 16 | 1024 | 1024 |
| | Github | 202M | 470M | 16 | 1024 | 1024 |
| Transformer (Window Attention) | PG-19 | 227M | 529M | 32 | 768 | 4608 |
| | arXiv | 202M | 470M | 16 | 1024 | 6144 |
| | Github | 202M | 470M | 16 | 1024 | 8192 |
| MegaByte | PG-19 | 201M+51M | 219M | 16/16 | 1024/512 | 4096 |
| | arXiv | 201M+51M | 219M | 16/16 | 1024/512 | 4096 |
| | Github | 201M+51M | 219M | 16/16 | 1024/512 | 4096 |
| SpaceByte (fixed $P$) | PG-19 | 201M+113M | 343M | 16/16 | 1024/768 | 6144 |
| | arXiv | 201M+113M | 343M | 16/16 | 1024/768 | 6144 |
| | Github | 201M+113M | 323M | 16/16 | 1024/768 | 8192 |
| SpaceByte | PG-19 | 201M+50M | 196M | 16/16 | 1024/512 | 6144 |
| | arXiv | 201M+50M | 196M | 16/16 | 1024/512 | 6144 |
| | Github | 151M+38M | 132M | 12/12 | 1024/512 | 8192 |

Table 4: **Model hyperparameters.** Hyperparameters for models shown in Table 2. In order to roughly match the FLOPs-per-byte of the other models, for the subword-level Transformer-1B, we used 40, 44, 37, and 31 layers for the PG-19, Stories, arXiv, and Github datasets, respectively.

| Model | Parameters | FLOPs-per-byte | $L$ $(L_\text{global}/L_\text{local})$ | $D$ $(D/D_\text{local})$ | Others |
|---|---|---|---|---|---|
| Transformer | 1B | $\approx 730$M | 40, 44, 37, or 31 | 1536 | subword-level |
| Transformer | 320M [7] | 732M | 22 | 1024 | byte-level |
| PerceiverAR | 248M [7] | | 17 | 1024 | latents $= 1024$ |
| MegaByte | 758M+262M [7] | 729M | 14/18 | 2048/1024 | $P = 8$ |
| MambaByte | 353M [6] | 713M | 53 | 1024 | $n_\text{state} = 16$ |
| SpaceByte | 793M+184M | 728M | 28/26 | 1536/768 | $T_\text{global} = 1344$ |

# A  Model Details

Hyperparameters for models shown in Table 1 and Table 2 are summarized in Table 3 and Table 4, respectively. In Figure 4, we show another perspective of Figure 3 where we plot the bits-per-byte vs the bytes trained divided by the number of model parameters.

For the subword models (but not the byte-level models), we tie the input embedding weights with the output linear matrix weights [54]. In self-attention layers, we use a key dimension equal to 64. Although we apply RoPE embeddings, we also included trained position embeddings.

**SpaceByte** For SpaceByte, we include trained position embeddings just before the first local transformer block, and just before the first global transformer block.

Table 5: **Model non-embedding parameter counts.** For MegaByte and SpaceByte, we separate the number of parameters ($m$) into the global ($m_{\text{global}}$) and local ($m_{\text{local}}$) model contributions. We ignore embeddings and subleading parameters, such as layer norms, but include de-embedding parameters. See Section A.1 for symbol definitions.

| Architecture | Parameters (non-embedding) | Component |
|---|---|---|
| Transformer | $m = L \times 4D^2$ 
 $+\ L \times 2e_{\text{ff}}D^2$ 
 $+\ DV$ | Multi-head attention 
 Feed-forward 
 De-embedding |
| MegaByte | $m_{\text{global}} = L_{\text{global}} \times 4D^2$ 
 $+\ L_{\text{global}} \times 2e_{\text{ff}}D^2$ 
 $m_{\text{local}} = D_{\text{local}}\frac{D}{P}$ 
 $+\ L_{\text{local}} \times 4D_{\text{local}}^2$ 
 $+\ L_{\text{local}} \times 2e_{\text{ff}}D_{\text{local}}^2$ 
 $+\ D_{\text{local}}V$ | Global multi-head attention 
 Global feed-forward 
 Global-to-local projection 
 Local multi-head attention 
 Local feed-forward 
 De-embedding |
| SpaceByte | $m_{\text{global}} = L_{\text{global}} \times 4D^2$ 
 $+\ L_{\text{global}} \times 2e_{\text{ff}}D^2$ 
 $m_{\text{local}} = L_{\text{local}} \times 4D_{\text{local}}^2$ 
 $+\ L_{\text{local}} \times 2e_{\text{ff}}D_{\text{local}}^2$ 
 $+\ D_{\text{local}}V$ | Global multi-head attention 
 Global feed-forward 
 Local multi-head attention 
 Local feed-forward 
 De-embedding |

Table 6: **Inference FLOPs-per-token.** We calculate the inference FLOPs-per-token in terms of the numbers of parameters ($m$), shown in Table 5. See Section A.1 for symbol definitions.

| Architecture | inference FLOPs-per-token |
|---|---|
| Transformer | $2m + 2L\left(2WD\right)$ |
| MegaByte | $2m_{\text{global}}\frac{1}{P} + 2L_{\text{global}}\left(2\frac{T}{P}D\right)\frac{1}{P} +$ 
 $2m_{\text{local}} + 2L_{\text{local}}\left(2PD_{\text{local}}\right)$ |
| SpaceByte | $2m_{\text{global}}\frac{T_{\text{global}}}{T_{\text{local}}} + 2L_{\text{global}}\left(2T_{\text{global}}D\right)\frac{T_{\text{global}}}{T_{\text{local}}} +$ 
 $2m_{\text{local}} + 2L_{\text{local}}\left(2W_{\text{local}}D_{\text{local}}\right)$ |

Just like the other models, SpaceByte is trained using a fixed context size of $T$ bytes. At the same time, we also fix the maximum global context size of $T_{\text{global}}$ patches. However, the number of patches in a given context of $T$ bytes is usually not exactly equal to $T_{\text{global}}$. To handle this mismatch during training, if the number of patches from applying the patching rule (see e.g. Figure 2) to a context of $T$ bytes is greater $T_{\text{global}}$, then we simply ignore the bytes within these extra patches when calculating the cross entropy. Alternatively, if the number of patches is less than $T_{\text{global}}$, then we pad the activations for the global transformer blocks with zeroes and ignore the output of these global blocks. Thus, the input activations to the global blocks is always a tensor of same shape for each iteration. This discrepancy between the maximal global context size $T_{\text{global}}$ and the actual number of patches results in a small fracton of wasted compute during training, which we roughly minimize by roughly tuning $T/T_{\text{global}}$. See Appendix C for pseudocode.

During inference, the model must stop predicting tokens before either the max number of bytes ($T$) or the max number of patches ($T_{\text{global}}$) is reached.

## A.1 FLOPs

The inference FLOPs-per-byte is the number of FLOPs required to output a byte of text during inference. We calculate the FLOPs-per-byte as the FLOPs-per-token divided by the average number of bytes per token (which is equal to 1 for byte-level models).

The FLOPs-per-token is the number of FLOPs required to output a token of text during inference (or byte of text for byte-level models). The FLOPs-per-token for the various architectures is shown in Table 6.

**Notation** For all architectures, $T$ is the context length; $D$ is the model dimension (of the global model for SpaceByte and MegaByte); $e_{\text{eff}} = 4$ is the model dimension expansion factor for feed-forward layers; and $V$ is the vocabulary size (which is 256 for byte-level models and 50257 for our subword models). For the transformer architecture, $L$ is the number of transformer blocks, and $W$ is the attention window size (which is equal to $T$ if window attention is not used). For SpaceByte and MegaByte, $D_{\text{local}}$ is the dimension of the local model; $L_{\text{local}}$ is the number of local transformer blocks; and $L_{\text{global}}$ is the number of global transformer blocks. For SpaceByte, $T_{\text{global}}$ is the maximum context size for the global model, and $W_{\text{local}}$ (which we set to $D_{\text{local}}$) is the attention window size for the local blocks. For MegaByte, $P$ is the patch size.

## B  Training Details

### B.1  Data

Each dataset prepared by downloaded it from Hugging Face[8], concatenating sequences together, and separating sequences with a special BOS token. When preparing a training sample with context size $T$, we uniformly and randomly sample a sub-sequence of length $T$ from the concatenated dataset. If a BOS token is found in this subset, we align the context with the first BOS token found; i.e. we take the context to be the first BOS token followed by the next $T - 1$ tokens in the concatenated dataset. If a BOS token is not found in the subset, we prepend a BOS token to the context. The context window is always full and always begins with a BOS token.

For the SpaceByte models, we always insert global blocks after a BOS token. A valid UTF-8 encoding never makes use of the byte values 254 or 255. We use 255 to encode the BOS token.

We train the SentencePiece tokenizers using the following command:
```
spm_train --input=train.txt --model_prefix=sp --model_type=bpe
    --vocab_size=50257 --num_threads=32 --byte_fallback=True
    --allow_whitespace_only_pieces=True --remove_extra_whitespaces=False
    --normalization_rule_name=identity --input_sentence_size=10000000
```

### B.2  Training

All models are trained using AdamW [55] with $\beta_1 = 0.9$, $\beta_2 = 0.98$, batch size 64, weight decay of 0.01, and gradient clipping [56] with a maximum norm of 1.0. Trainable parameters are randomly initialized using a normal distribution with standard deviation $\sigma_{\text{init}} = 1$ for all parameters except for linear weight matrices, which are initialized with standard deviation of $\sigma_{\text{init}} = 1/\sqrt{d_{\text{in}}}$, where $d_{\text{in}}$ is the input dimension for the linear layer. We scale the learning rate for each parameter by its initialization standard deviation $\sigma_{\text{init}}$.

With this setup, we found in our early hyperparameter search experiments that the optimal max learning rate for all models is approximately $\gamma = 0.005B^{-0.5} = 0.000625$, where $B = 64$ is the batch size. We therefore used $\gamma = 0.000625$ as the max learning rate for all models trained in this work. We applied a linear learning rate warmup over the first 1% of training iterations. We also multiply the learning rate by a "half-cosine" learning rate decay function $\cos(\pi x/2)$, where $0 \le x \le 1$ is the fraction of training iterations completed.[9]

Each model was trained using PyTorch on a single 40GB Nvidia A40 and A100 GPUs with mixed-precision (bfloat16 and float32) training and FlashAttention [57, 58]. SpaceByte-793M+184M took the longest to train, requiring about 10 days on an A100 GPU.[10]

---

[8]https://huggingface.co/datasets/pg19
https://huggingface.co/datasets/lucadiliello/STORIES
https://huggingface.co/datasets/monology/pile-uncopyrighted

[9]In our setup, we found $\cos(\pi x/2)$ to slightly outperform the more standard cosine decay from 1 to 0.1.

[10]We very roughly estimate that additional preliminary and failed experiments not shown in this work required roughly as many FLOPs as the experiments shown in this work.

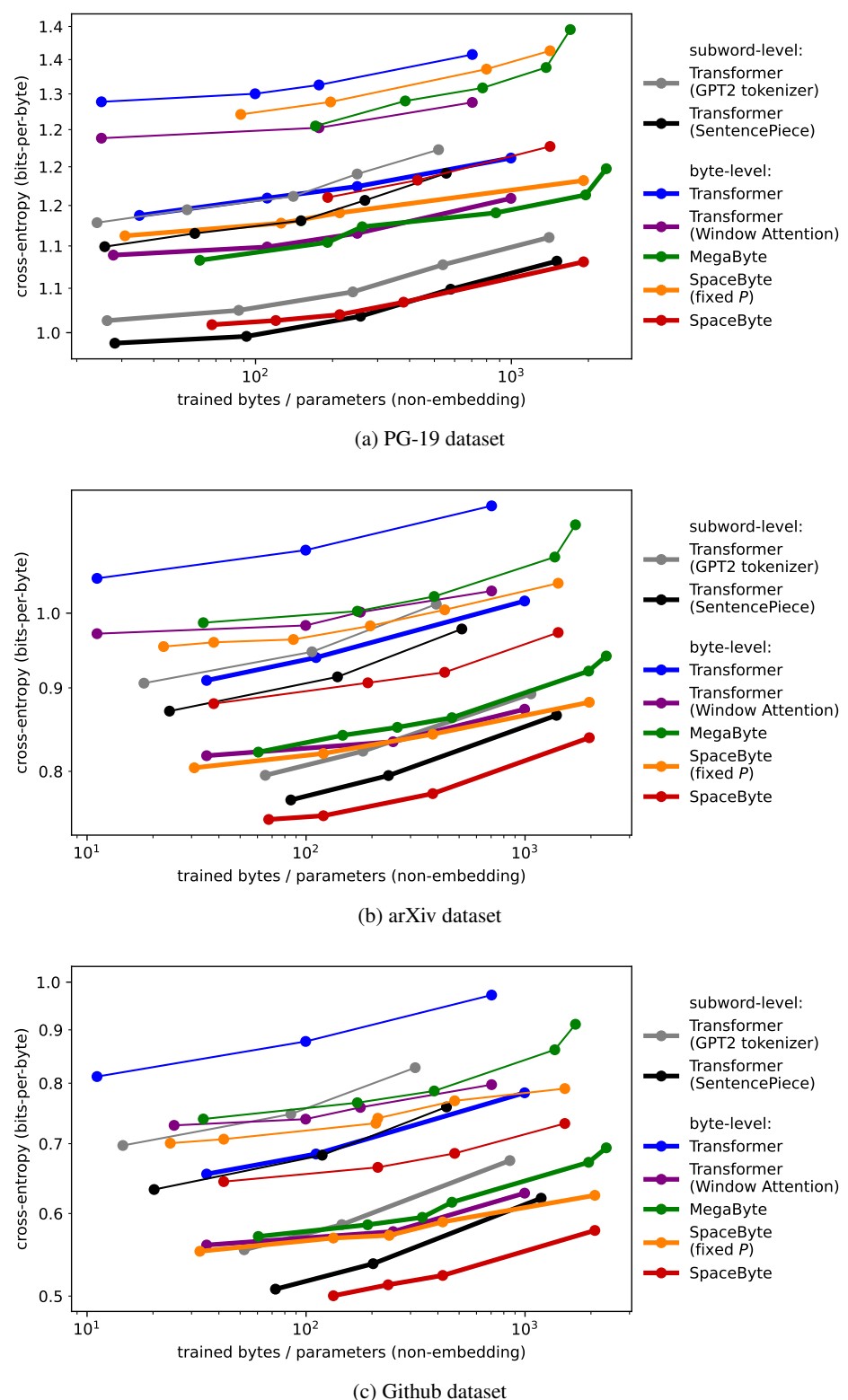

(a) PG-19 dataset

(b) arXiv dataset

(c) Github dataset

Figure 4: The Pareto frontier models from Figure 3, where we plot the bits-per-byte vs the number of bytes used for training divided by the number of non-embedding parameters (defined in Table 5).

### B.3 Hyperparameter Grid

We train models using a grid of different model dimensions and numbers of layers. In our early small-scale experiments, we found that the hyperparameter grid described below effectively explores the bits-per-byte and FLOPs-per-byte Pareto frontier for all models. To simplify the hyperparameter grid, we restrict ourselves to model dimensions and layer numbers to half-powers of two, i.e. a power of two times 1 or $\frac{3}{2}$.

For models trained using $10^{18}$ FLOPs, we train model dimensions $D \in \{384, 512, 768\}$. For models trained using $10^{19}$ FLOPs, we train model dimensions $D \in \{512, 768, 1024\}$.

For SpaceByte and MegaByte, $D$ is the global model dimension. The local model dimension is chosen from $D_{\text{local}} \in \{\frac{1}{2}D, \frac{3}{4}D\}$ if $D$ is a power of two, or $D_{\text{local}} \in \{\frac{1}{2}D, \frac{2}{3}D\}$ if $D$ is a power of two times $\frac{3}{2}$. However, in order to avoid excessively low FLOP utilization, we restrict $D_{\text{local}} \geq 256$ (or $D_{\text{local}} \geq 384$) for models trained using $10^{18}$ FLOPs (or $10^{19}$ FLOPs).

To set the number of layers, we roughly follow Levine et al. [59], who found that the compute-optimal number of layers for a Transformer roughly follows $L \sim 12.5 \log_2(D/154)$. We round this number to the nearest half-power of two to obtain $L_D$, for which $L_{384} = 16$, $L_{512} = 24$, $L_{768} = 32$, and $L_{1024} = 32$. For Transformer models, we choose the number of layers from $L \in \{\frac{1}{2}L_D, L_D\}$.

For SpaceByte and MegaByte models, we choose the number of local and global layers from $L_{\text{local}} = L_{\text{global}} \in \{\frac{3}{8}L_D, \frac{1}{2}L_D\}$ if $L_D$ is a power of two, or $L_{\text{local}} = L_{\text{global}} \in \{\frac{1}{3}L_D, \frac{1}{2}L_D\}$ if $L_D$ is a power of two times $\frac{3}{2}$.

## C  Pseudocode

See Listing 1 for Pytorch pseudocode for the SpaceByte forward method. The implementation of SpaceByte that we used in our experiments can be found at github.com/kjslag/spacebyte.

Listing 1: Pytorch pseudocode for SpaceByte

```python
def forward(self, tokens, targets=None):
    B, T = tokens.shape # batch size, context size
    T_global = self.global_context_size
    D_local = self.local_model_dimension
    D = self.global_model_dimension

    # embedding:
    x = self.token_embedding(tokens)
    x = x + self.local_position_encoding
    # initial local transformer blocks:
    for block in self.initial_blocks:
        x = block(x)

    # global block insertion rule:
    use_global = ( # not a letter, number, or UTF-8 continuation byte
        (tokens < ord('0')) |
        ((ord('9') < tokens) & (tokens < ord('A'))) |
        ((ord('Z') < tokens) & (tokens < ord('a'))) |
        ((ord('z') < tokens) & (tokens < 0b1000_0000)) |
        (0b1100_0000 <= tokens) )
    use_global[:, 1:] &= use_global[:, :-1].bitwise_not() # not
        preceded by another spacelike byte
    use_global |= tokens == self.BOS_token # always use global blocks
        after BOS tokens

    # calculate global block indices:
    num_global = torch.full((B,), -1) # number of global blocks used
    global_idx = torch.full((B, T_global), T-1) # global block indices
    for b in range(B):
        idx, = use_global[b].nonzero(as_tuple=True)
        if targets is not None and len(idx) > T_global:
            # ignore targets with insufficient global blocks:
            targets[b, idx[T_global]:] = -1
        num_global[b] = len(idx[:T_global])
        global_idx[b, :num_global[b]] = idx[:T_global]

    # select activations for global blocks:
    y = x.gather(1, global_idx[:,:,None].expand(B, T_global, D_local))
    # expand model dimension by padding with zeros:
    y = torch.cat([torch.zeros(B, T_global, D - D_local), y], -1)

    # global transformer blocks:
    y = y + self.global_position_encoding
    for block in self.global_blocks:
        y = block(y)

    # add global block activations to local blocks:
    x = torch.stack([
        x[b].index_add(0, global_idx[b, :n], y[b, :n, -D_local:])
        for b, n in enumerate(num_global) ])

    # final local transformer blocks:
    for block in self.final_blocks:
        x = block(x)
    # de-embedding:
    logits = self.logits_linear(self.layer_norm(x))

    cross_entropy_loss = None
    if targets is not None:
        cross_entropy_loss = torch.nn.functional.cross_entropy(
            logits.view(B*T, 256), targets.view(B*T),
            ignore_index=-1).view(B, T)
    return logits, cross_entropy_loss
```

