# OpenReview forum: "SpaceByte: Towards Deleting Tokenization from Large Language Modeling"
_NeurIPS.cc/2024/Conference — NeurIPS 2024 poster_

### Official Review · Reviewer_ZB8e · 2024-07-05

**Soundness:** 4
**Presentation:** 4
**Contribution:** 3
**Rating:** 7
**Confidence:** 4

**Summary:**

The authors propose a byte-level architecture called SpaceByte that involves local blocks (lower dimension, windowed attention) and global blocks (higher dimension, global attention) where the global blocks are between chunks of local blocks and only selectively applied. The global blocks are applied to "spacelike" characters (such as an actual space character), with the intuition being that at such points predicting the next character (like start of word) would be harder. The authors run extensive experiments to show that their proposed architecture outperforms previous byte-level architectures and in fact performs similarly to subword-level transformer baselines.

**Strengths:**

**Originality**: While this work seems to share much in common with past byte-level architecture work, the intuitive idea of some bytes being harder to predict (in a rule predictable way) is very natural and useful (and first presented in this work, as far as I am aware). The architecture itself is also novel.

**Quality**: The authors run a lot of experiments to demonstrate the capabilities of their architecture.

**Clarity**: The paper was clear, down-to-earth, and easy to understand.

**Significance**: This paper shows that byte-level architectures can be competitive with subword tokenization transformers in a flop-matched setting.

**Weaknesses:**

* As a main weakness I'd say that even though this method beats other byte-level approaches and is comparable to subword tokenization, I don't see a compelling reason to use it over standard tokenization. In the Introduction the authors mention several downsides of tokenization, but don't have results that focus on these issues per se. I'm also a bit confused about the "additional modeling complexity (line 20)" point. It seems like tokenization is actually simpler.
* It would have been nice to see some ablations on the architecture (for example, the local -> global -> local layer ordering seems a little arbitrary).
* It would have been nice to see some modifications of the global rule. For example, only "space" characters like actual space and newline vs the more broad definition of "space" used in the paper.
* It might be nice to include more info on what the "spacelike" tokens actually look like (what are the most common "spacelike" tokens and what percentage of all "spacelike" tokens do these make up).

**Questions:**

Please see weaknesses for my questions.

**Limitations:**

Yes. The paper does a good job of this.

---

> ### Author Rebuttal · Authors · 2024-08-03
>
> Thank you very much for your insightful and thorough review.
>
> ## Weakness 1
> We agree that there is not a compelling case to use SpaceByte over a subword Transformer. However, we hope that iterating upon the SpaceByte architecture could eventually yield a compelling byte-level model. And we believe that SpaceByte is a significant milestone in this process since it's the first attention-based architecture to achieve performance parity with subword Transformers.
>
> We would have loved to address the other tokenization issues, but we unfortunately ran out of time. However, we were very pleased to see that MambaByte [7] (i.e. byte-level Mamba [25]) exhibits a significantly higher character-level noise-tolerance than subword Transformers. We expect SpaceByte to show a similar tolerance improvement, although we have unfortunately not had time to experimentally check this.
>
> The sentence on line 20 is not making a claim about SpaceByte. We simply mean that additional modeling complexity is a disadvantage of tokenized Transformers over e.g. a byte-level Transformer. We agree that the extra complexity of SpaceByte largely cancels out the simplicity of removing the tokenizer, and thus simplicity is unfortunately not a compelling reason to use SpaceByte over a tokenized Transformer.
>
> ## Weakness 2
> We apologize for the lack of ablations. We don't have ablations because when faced with an architecture decision, we simply made the simplest choice that should work well. Examples include: the local block attention window (else the FLOPs for the local attention would be roughly 36x larger and dominate the model's flops), the lack of a linear layer to change model dimensions between local and global blocks (which MegaByte had but we found in preliminary experiments to slightly hurt FLOP-controlled performance), and our choice of spacelike characters.
>
> The local -> global -> local ordering is not at all arbitrary. If no local layers preceded the global layers (e.g. global -> local), then the global layers would not have any information about the majority of the bytes. Similarly, if no local layers followed the global layers (e.g. local -> global), then the global layers would have no influence on the majority of the output bytes. As such, local -> global -> local is the simplest good choice.
>
> ## Weakness 3
> We studied something like this in preliminary experiments, and found that it performed slightly worse on the datasets we studied.
>
> ## Weakness 4
> That is a nice suggestion, although it is certainly very dataset-dependent. We may include it in the next version. Until then, we hope that the demonstration in Figure 2 is helpful.

---

### Official Review · Reviewer_zbbv · 2024-07-07

**Soundness:** 3
**Presentation:** 3
**Contribution:** 2
**Rating:** 5
**Confidence:** 4

**Summary:**

The paper introduces SpaceByte, a byte-level Transformer model that incorporates larger transformer blocks at specific byte boundaries to enhance performance in language modeling tasks. While the approach bears similarities to previous work (e.g. MegaByte), it demonstrates improved performance over traditional tokenization models. The study focuses on specific languages, acknowledging limitations in generalizability, particularly in languages like Chinese, which do not use spaces between words.

**Strengths:**

1. The paper presents a novel approach in utilizing byte-level Transformer models with specific block insertions to enhance language modeling performance.

2. The study provides insights into the limitations of tokenization models and the potential benefits of byte-level architectures.

3. The experimental methodology is well-documented, controlling for compute costs and providing detailed training details in the appendices.

**Weaknesses:**

1. The novelty of the SpaceByte approach may be limited by similarities to previous models.

2. The study's focus on specific languages, without detailed experiments on broader language datasets, limits the generalizability of the findings.

3. The paper could benefit from a more extensive discussion on the unique contributions of SpaceByte compared to existing models.

**Questions:**

1. How do the authors plan to address the limitations in generalizability to languages beyond the ones studied in the current work?

2. Are there plans to conduct experiments on a more diverse set of language datasets to further validate the effectiveness of SpaceByte in different linguistic contexts?

**Limitations:**

The authors have acknowledged the limitations of their work, particularly in terms of language generalizability. To enhance the impact of the study, it would be beneficial for the authors to consider conducting experiments on a broader range of language datasets to strengthen the validity and applicability of SpaceByte across various linguistic domains.

---

> ### Author Rebuttal · Authors · 2024-08-03
>
> Thank you very much for your careful review of our work and for identifying its key strengths.
>
> ## Weakness 3
> As discussed in Section 3, our primary contributions beyond prior works is to show how to scale word-boundary byte-level modeling to more diverse text modalities while roughly matching the performance of subword-level models in compute-controlled experiments. Previous works that studied word boundary LLMs did not demonstrate that this is possible or how to do it.
>
> ## Question 1
> This is a very important direction for future work. Unfortunately, we will not have time to address it in the foreseeable future. However, Reference 9 studied some promising techniques, such as using entropy to partition the patch boundaries. It would be useful to see if this approach is also capable of matching subword Transformer performance, especially on broader language datasets.
>
> ## Question 2
> We unfortunately do not currently have plans for this (due to time limitations).

---

### Official Review · Reviewer_HARv · 2024-07-13

**Soundness:** 3
**Presentation:** 2
**Contribution:** 2
**Rating:** 6
**Confidence:** 4

**Summary:**

Byte-level modeling allows transformers to circumvent subword tokenization, thus avoiding the many weaknesses introduced by tokenization. However those models are not very performant compared to subword-based transformers. This paper proposes a novel architecture named *SpaceByte*. The idea is to add an extra wide layer (i.e. "global block") between regular transformer layers (i.e. "local block") if and only if "the byte does not encode a letter, number, or UTF-8 continuation byte". Results show that when normalizing for training FLOPs, SpaceByte achieves the best PPL on multiple datasets. Plus, at different model dimensions and number of layers (?), SpaceByte always achieve the optimal or near-optimal Pareto optimality between PPL and inference FLOPs-per-byte.

**Strengths:**

1. The idea is simple and well-motivated
2. Experiment results show that it is competitive when compared subword-based models

**Weaknesses:**

1. While normalizing for FLOPs, I think there is some potential that the proposal will make the transformer blocks significantly harder to batch (due to the input-dependent dynamic structure of the model), hence the clock-time of the inference might actually be slower with the same FLOPs. It would be good to see some discussions on that.
2. There is no ablations as to whether the global blocks actually need larger dimension.
3. Presentation of the paper could be improved. See my question/suggestion in the next section.

**Questions:**

### Presentation Suggestions

1. It's worth explaining more about how context lengths work differently for global vs. local blocks in Section 2. I'm also not very sure why the larger context length is necessary.
2. I'd suggest switching Section 2 and 3 and talk about related work first.
3. Three suggestions for Figure 3 -- a. make it clear in the caption that you are changing inference FLOPs with different model dimensions & layers (per line 206); b. use dashed/solid lines rather than thin/thick lines; c. mark the pareto frontier for each subfigures.

### Questions

1. Is the FLOP limits you imposed enough to make the models converge adequately on those datasets? It might have been the case that your model simply converges faster, but if trained longer, doesn't work as well as, e.g. sub-word models (which weakens your case).
2. Why set the dimension and context length equal?
3. I'm very confused what's the difference between section 6 and 5. How does the results in section 6 further complements the one in section 5?
4. In Table 2, you only have one context size, but don't global blocks have larger context size? What context size is being reported then?

**Limitations:**

See weakness point 1. I'm relatively confident that this could be a potential limitation that needs to be addressed.

---

> ### Author Rebuttal · Authors · 2024-08-03
>
> Thank you for your detailed review of our work. Although we agree that SpaceByte is more challenging to batch than a Transformer, we would like to emphasize that our work does establish a very significant milestone in byte-level LLMs because SpaceByte is the first byte-level attention-based architecture to demonstrate performance parity with the subword Transformer architecture. We believe this significance warrants acceptance (while leaving room for future work to show that the batching issue is not a significant hurdle).
>
> ## Weakness 1
> You are correct that batching during inference is more challenging with SpaceByte. Nevertheless, we do believe that with a moderate amount of effort, a high FLOP utilization could be maintained with batching and without significantly sacrificing latency. One solution would be to maintain three separate queues for three parts of the model pipeline:
> 1. embedding + initial local blocks
> 2. global blocks
> 3. final global blocks + de-embedding
>
> Each queue processes a batch of input tokens or activations at a time. The batch size and context lengths are variable. Variable context lengths is not an issue since padding could be used (and padding is already typically used for batching due to variable prompt lengths). In a multi-GPU setup, it could be useful to put different queues on different GPUs in a 1-2-1 ratio for the three queues (to roughly even out the FLOPs). And 8 GPU setup would then naturally allow for two instances of each queue, which would be useful for reducing latency due to waiting in a queue. Thus, although the batching issue is annoying, we do not think that it is a major obstacle.
>
> ## Weakness 2
> In preliminary experiments, we found that taking equal model dimensions for the local and global blocks to be significantly worse. Therefore, in our hyperparameter grid search, we take the local block dimension to be:
> D_local = 1/2 * D or (3/4 if D is a power of two else 2/3) * D
> where D is the global block dimension. This is specified in Appendix B.3. But as seen in Tables 3 and 4, which shows optimal hyperparameters found by our grid search, the smaller D_local = 1/2 * D performs better. Therefore, our paper does show strong evidence that it's important to use a larger model dimension for the global blocks.
>
> ## Presentation Suggestion 1
> Thank you for the suggestion. We have appended the following sentence to line 76 at the end of Section 2: "The global blocks use a global attention that attends to all other global blocks."
>
> The context length that we experiment on is the natural (and roughly optimal) choice for all architectures that we evaluated. In particular, the chosen context length is natural and roughly optimal for the subword tokenizer since it roughly balances the FLOPs between the attention and MLP layers. SpaceByte uses a similar context length (after converting tokens to bytes) for a fair comparison (see Table 3 for precise numbers). But this context length would be too large for the local blocks to be efficient if we did not utilize an attention window since then the local attention blocks would use significantly more FLOPs than the local MLP blocks. An attention window prevents this inefficiency.
>
> ## Presentation Suggestion 2
> Thank you for the suggestion. We made the choice to put the Related Work section after the SpaceByte section so that we could comment on how SpaceByte relates to the related works.
>
> ## Presentation Suggestion 3a
> Thank you very much for this suggestion! We agree that it greatly improves the clarity of the caption. We added the following additional sentence in the new version:
> "Each dot describes a model with a different number of layers and/or model dimension."
>
> ## Presentation Suggestion 3b
> Thank you for the suggestion. We tried dashed/solid lines as per your suggestion; but we feel the thin/thick lines are cleaner and easier to read.
>
> ## Presentation Suggestion 3c
> We already show the Pareto frontier, which is drawn for each model using a thin or think line.
>
> ## Question 1
> It depends on what you mean by "adequately." The models are certainly not trained to convergence, as that would be prohibitively expensive in practice. In practice, one (almost always) does not care how well a LLM performs when trained to convergence. Instead, one cares about how well a LLM performs for a given cost budget. There are two important budgets that we consider: a training budget and an inference budget. We use training and inference FLOPs as a simple proxy for these respective costs since FLOPs are independent of software and hardware choices. This results in the Pareto frontiers shown in Figure 3. In this sense, all of the models shown in Figure 3 are very adequately trained as they lie on the Pareto frontier of performance vs cost.
>
> ## Question 2
> We set the model dimension and context length equal so that the FLOPs are roughly balanced between the MLP and attention blocks, which tends to be the most efficient. This was a simple choice that we thought made the most sense for a fair comparison that evaluates all of the models at their best.
>
> ## Question 3
> Section 5 is our main experiment. But we include Section 6 to make closer comparisons to the MegaByte and MambaByte experiments. Table 2 of Section 6 also includes PerceiverAR and MambaByte, which were not studied in Section 5. Furthermore, Table 2 shows MegaByte results trained by the MegaByte authors to help demonstrate that our finding that SpaceByte outperforms MegaByte is not just because we didn't train MegaByte well. (Unfortunately, this fact is only established for the PG-19 and Stories datasets in Table 2, since MegaByte was trained and evaluated on proprietary arXiv and Github datasets that are slightly different than the public datasets that we have access to).
>
> ## Question 4
> We report the context size available to the LLM. For SpaceByte, this is the total number of bytes in the input sequence.

---

> > ### Comment · Reviewer_HARv · 2024-08-12
> > **Post-rebuttal Comment**
> >
> > Thank you for the very detailed rebuttal. It clarified my main confusions during my first read of the paper.
> >
> > Re: Question 1 -- By "adequately" I mean train until convergence, but your budget argument is valid and I will take it.
> >
> > I still have reservations on the practicality of the proposed method (esp. batching), but otherwise I think the paper should be accepted. I'm improving my final evaluation to "weak accept".
> >
> > Since the authors also agree that batching is challenging with this proposal, please make sure you make some space to address  this issue/limitation in the final draft of the paper.

---

### Official Review · Reviewer_8HNC · 2024-07-16

**Soundness:** 4
**Presentation:** 4
**Contribution:** 3
**Rating:** 8
**Confidence:** 4

**Summary:**

This paper proposes SpaceByte, a byte-level decoder architecture for language modeling. As opposed to comparable models such as MegaByte, SpaceByte applies global transformer blocks after space-like characters, not after patches of a fixed size. The authors show that this approach leads to a substantially improved performance: compared to several other byte-level decoder architectures (including MegaByte), SpaceByte is the only one that matches or even exceeds the performance of subword-level models trained with the same compute budget.

**Strengths:**

The proposed SpaceByte architecture is novel. The experimental setup is rigorous --- I think the authors did a great job in (a) evaluating a range of different architectures and (b) ensuring a fair comparison by controlling compute costs and using bits-per-byte as the evaluation measure. The results show clear performance improvements for SpaceByte, highlighting its advantages compared to other byte-level decoder architectures. Overall, I liked the paper very much and think that it should be accepted.

**Weaknesses:**

The main weakness that I see is that the authors only evaluate the different architectures using bits-per-byte, not any downstream task. To become a real alternative to subword-level models in practice, it would be important to show that the similar language modeling performance translates to a similar downstream task performance. While the authors cite work indicating that this might be the case (Huang et al., 2024), without actual experiments it is unclear whether this holds for the examined architectures as well.

**Questions:**

Is there any specific reason you did not include evaluations on downstream tasks?

**Limitations:**

Yes, the authors discuss limitations as part of the conclusion.

---

> ### Author Rebuttal · Authors · 2024-08-03
>
> Thank you very much for your thoughtful review and for accurately assessing strengths and weaknesses of our work.
>
> We would have loved to include evaluations on downstream tasks, but we unfortunately ran out of manpower and time.

---

### Decision · Program_Chairs · 2024-09-25

**Decision:**

Accept (poster)

**Comment:**

Paper introduces SpaceByte, a novel byte-level language model designed to bridge the performance gap between byte-level and subword-based models.  The core idea revolves around strategically inserting wider, global transformer blocks after "space-like" bytes, aiming to improve performance on predicting word beginnings.

The notable weaknesses like limited generalizability, lack of downstream evals and ablations need to be addressed in the future and discussed in the paper. That said, the very topic this paper tackles and the novelty of the proposed method with the early wins potentially lead to acceptance as a poster.